# Inhibiting weld cracking in high-strength aluminium alloys

Yanan Hu[1,2], Shengchuan Wu [1,3] ✉, Yi Guo[4], Zhao Shen [5,6] ✉,
Alexander M. Korsunsky [7], Yukuang Yu[1], Xu Zhang [2], Yanan Fu[8], Zhigang Che[9],
Tiqiao Xiao[8], Sergio Lozano-Perez [6], Qingxi Yuan [10], Xiangli Zhong [3],
Xiaoqin Zeng[5], Guozheng Kang[1,2] & Philip J. Withers [3] ✉

Cracking from a fine equiaxed zone (FQZ), often just tens of microns across, plagues the welding of 7000 series aluminum alloys. Using a multiscale correlative methodology, from the millimeter scale to the nanoscale, we shed light on the strengthening mechanisms and the resulting intergranular failure at the FQZ. We show that intergranular AlCuMg phases give rise to cracking by microvoid nucleation and subsequent link-up due to the plastic incompatibility between the hard phases and soft (low precipitate density) grain interiors in the FQZ. To mitigate this, we propose a hybrid welding strategy exploiting laser beam oscillation and a pulsed magnetic field. This achieves a wavy and interrupted FQZ along with a higher precipitate density, thereby considerably increasing tensile strength over conventionally hybrid welded butt joints, and even friction stir welds.

Modern welding technology can trace its roots as far back as the latter half of the 19th century[1]. Nowadays, it is an everyday tool in the energy, shipbuilding, automotive, aircraft, aerospace, and railway industries. It enables the assembly of lightweight structures, which is of paramount importance in reducing energy consumption and carbon emissions[2]. In this respect, lightweight aluminum (Al) alloys have been increasingly deployed in recent decades. Use of high-strength Al alloys, such as aluminum–lithium (Al–Li) and 7000-series (Al–Zn–Mg–Cu) alloys, in particular, has become increasingly widespread[2,3]. One longstanding challenge is to overcome the local softening and cracking issues associated with conventional welding. This has seriously hampered long-term service applications[4–6] and led to a focus on solid state friction stir welding[7]. In many cases, this strength reduction is associated with the so-called fine equiaxed zone (FQZ) prevalent in fusion welds of these materials. At the microscopic level, the unique microstructural features associated with the FQZ are key because the

precipitation characteristics have a considerable influence on the mechanical properties and failure behavior. FQZs have also been observed in the welded joints of other Al alloys and steels (see Supplementary Table 1). To date, issues associated with FQZs have not received the attention that they deserve from a structural integrity point of view and a better understanding of the softening and failure mechanisms related to the FQZ is required.

The formation of the FQZ has been well studied in terms of the alloy constituents, solidification, base materials (BM), welding parameters, thermal history, and molten pool dynamics[6,8]. For Al–Li and Zr-containing Al alloys, the FQZ is formed through heterogeneous grain nucleation aided by $Al_3(Li_x, Zr_{1-x})$ and $Al_3Zr$ respectively[6]. However, the details of the softening and cracking behavior of the FQZ have not yet been clearly elucidated. This may be attributed to the fact that the FQZ is very narrow, containing very fine equiaxed grains posing a challenge to the precise characterization of the microstructure, properties, and

[1]State Key Laboratory of Traction Power, Southwest Jiaotong University, Chengdu, PR China. [2]School of Mechanics and Aerospace Engineering, Southwest Jiaotong University, Chengdu, PR China. [3]Henry Royce Institute, Department of Materials, The University of Manchester, Manchester, UK. [4]Institute of Metal Research, Chinese Academy of Sciences, Shenyang, PR China. [5]School of Materials Science and Engineering, Shanghai Jiao Tong University, Shanghai, PR China. [6]Department of Materials, University of Oxford, Oxford, UK. [7]Department of Engineering Science, University of Oxford, Oxford, UK. [8]Shanghai Synchrotron Radiation Facility (SSRF), Shanghai Advanced of Sciences, Shanghai, PR China. [9]Science and Technology on Power Beam Processes Laboratory, AVIC Manufacturing Technology Institute, Beijing, PR China. [10]Beijing Synchrotron Radiation Facility (BSRF), Chinese Academy of Sciences, Beijing, PR China. ✉e-mail: wusc@swjtu.edu.cn; shenzhao081@sjtu.edu.cn; p.j.withers@manchester.ac.uk

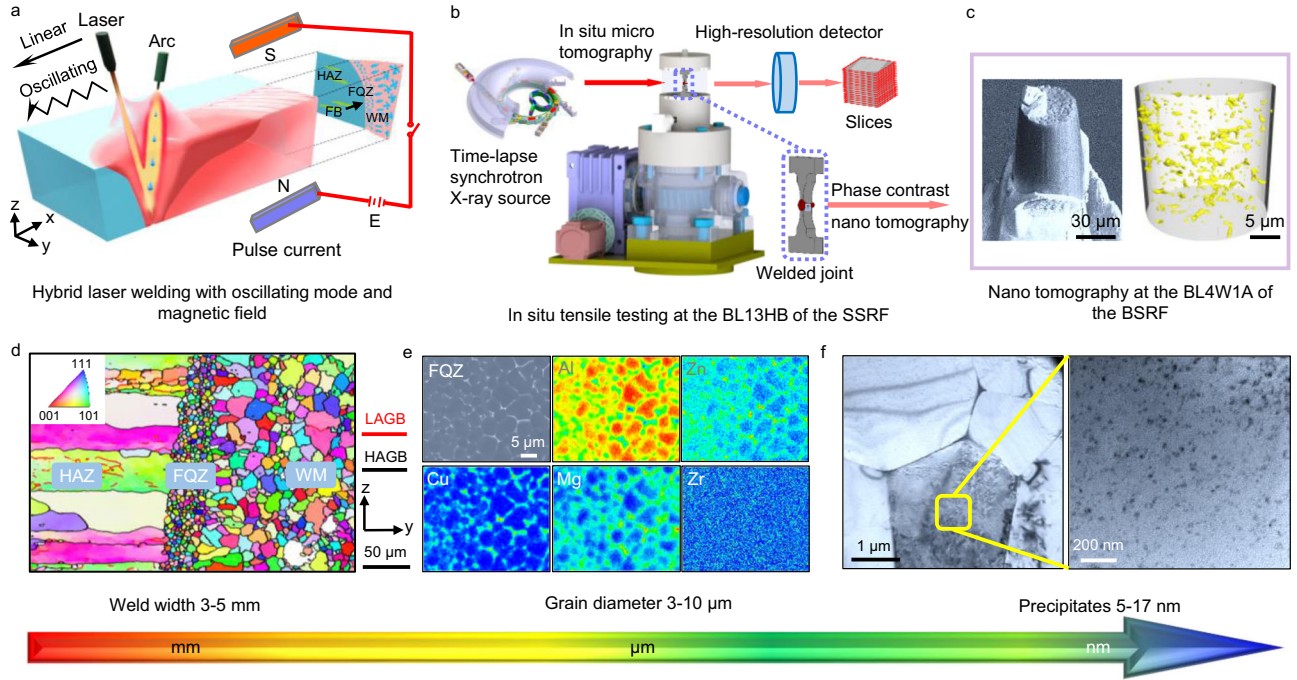

**Fig. 1 | Multiscale characterization of the fine equiaxed zone (FQZ). a** Schematic of the welding process where the HAZ, WM and FB represent heat affected zone, weld metal and fusion boundary, respectively. **b** Schematic of in situ tensile synchrotron radiation X-ray micro computed tomography (microCT) at the 13HB beam line (BL13HB) of the Shanghai Synchrotron Radiation Facility (SSRF). **c** High-resolution synchrotron X-ray nanoCT at the 4W1A beam line (BL14W1A) of the Beijing Synchrotron Radiation Facility (BSRF). **d** Electron backscatter diffraction (EBSD) inverse-pole figure (IPF) map across the fusion boundary (HAZ (left); FQZ (centre), weld (right)) where the high-angle grain boundaries (HAGBs, >10°) and the low-angle grain boundaries (LAGBs, 5–10°) are colored black and red, respectively. **e** Electron probe micro analysis (EPMA) maps showing chemical constitutions of the intergranular phases. **f** Bright-field transmission electron microscopy (TEM) images of precipitates in the interior of FQZ grains at different magnifications.

the associated damage accumulation sequence. Further, the in-service performance largely depends on microstructural features across length scales ranging from the macro- to the nano-scale. Recently, there has been a focus on combining various imaging methods with different resolutions to link the micro- and nano-scale features through what is termed 'correlative characterization' approaches[9]. Here the behavior of the FQZ is interrogated across multiple length scales by multiscale correlative tomography[10–12] to shed light on the various damage evolution mechanisms and their sequence.

Here we consider the FQZ arising from a hybrid laser and arc butt weld (HLAW) of 7050 Al alloy (see Fig. 1a) bringing together multiple techniques to study the microstructure across the scales (Fig. 1b–f). The relatively high rates of crystal nucleation and the very high solidification rates near the fusion boundary (FB) give rise to a large number of fine non-dendritic equiaxed grains located between the heat affected zone (HAZ) and the central weld metal (WM)—see Fig. 1d. The narrow (50–100 μm wide) FQZ is frequently missed when mapping hardness across the weld zone by Vickers hardness testing. Here the grain orientation, size distribution, and grain boundary characteristics in the FQZ have been characterized by electron backscatter diffraction (EBSD). These grains exhibit a random crystal orientation and their equivalent diameters range from 3 to 10 μm (average size ~7 μm). Our EBSD measurements show that in the FQZ a large fraction (~87%) of the boundaries are high-angle grain boundaries (HAGBs, >10°). Electron probe micro analysis (EPMA) reveals that these grain boundaries are highly enriched by the strengthening elements Zn, Mg, and Cu due to segregation during solidification (Fig. 1e). As a result, the grain boundaries are decorated with interconnected phases. Furthermore, the combination of severe elemental segregation and fast cooling act to limit the reprecipitation of the precipitates within the grains. Transmission electron microscopy (TEM) clearly shows that the distribution of the precipitates in the FQZ (Fig. 1f) represents a volume

fraction of just ~0.2% and an average radius of ~11 nm, compared to ~3.7% and ~15 nm respectively in the BM.

In this work we focus our attention on identifying the critical microstructural aspects affecting the local softening and intergranular failure to better understand the effect of the FQZ on softening and failure. First, we estimate the contribution to the yield strength of the grain size, dislocation density, solute and precipitate strengthening mechanisms using classical strengthening models across the different regions of the weld. Second, we employ a multiscale correlative tomography procedure to investigate the damage evolution and intergranular failure behavior, through in situ synchrotron radiation X-ray micro computed tomography (SR-μCT) during tensile straining (Fig. 1b), high-resolution synchrotron X-ray nanoCT (Fig. 1c) and energy dispersive spectrometry (EDS) in the TEM. Finally, we have developed an effective strategy to mitigate the effects of the FQZ by oscillating the laser beam and applying a pulsed magnetic field to disturb the FQZ, leading to a higher tensile strength, even reaching the strength level obtained by solid state welding.

## Results

### Softening mechanism

Nanoindentation testing (Fig. 2a) shows the narrow FQZ to be, by some margin, the softest region across the weld zone (hardness ~54% that of the BM). To understand the origin of the softening, we first performed a quantitative analysis of the average precipitate characteristics (radius, $r$, and volume fraction, $f$), grain sizes ($d$), solute concentrations ($c_{Zn}$, $c_{Mg}$, and $c_{Cu}$), and dislocation densities ($\rho$) across the four regions of the weld (see below). The results are summarized in Table 1. Based on these values the various strengthening contributions (grain size strengthening ($\Delta\sigma_{gb}$), solid-solution strengthening ($\Delta\sigma_{ss}$), dislocation strengthening, ($\Delta\sigma_{dis}$), and precipitation strengthening ($\Delta\sigma_{ppt}$)) can be estimated by the

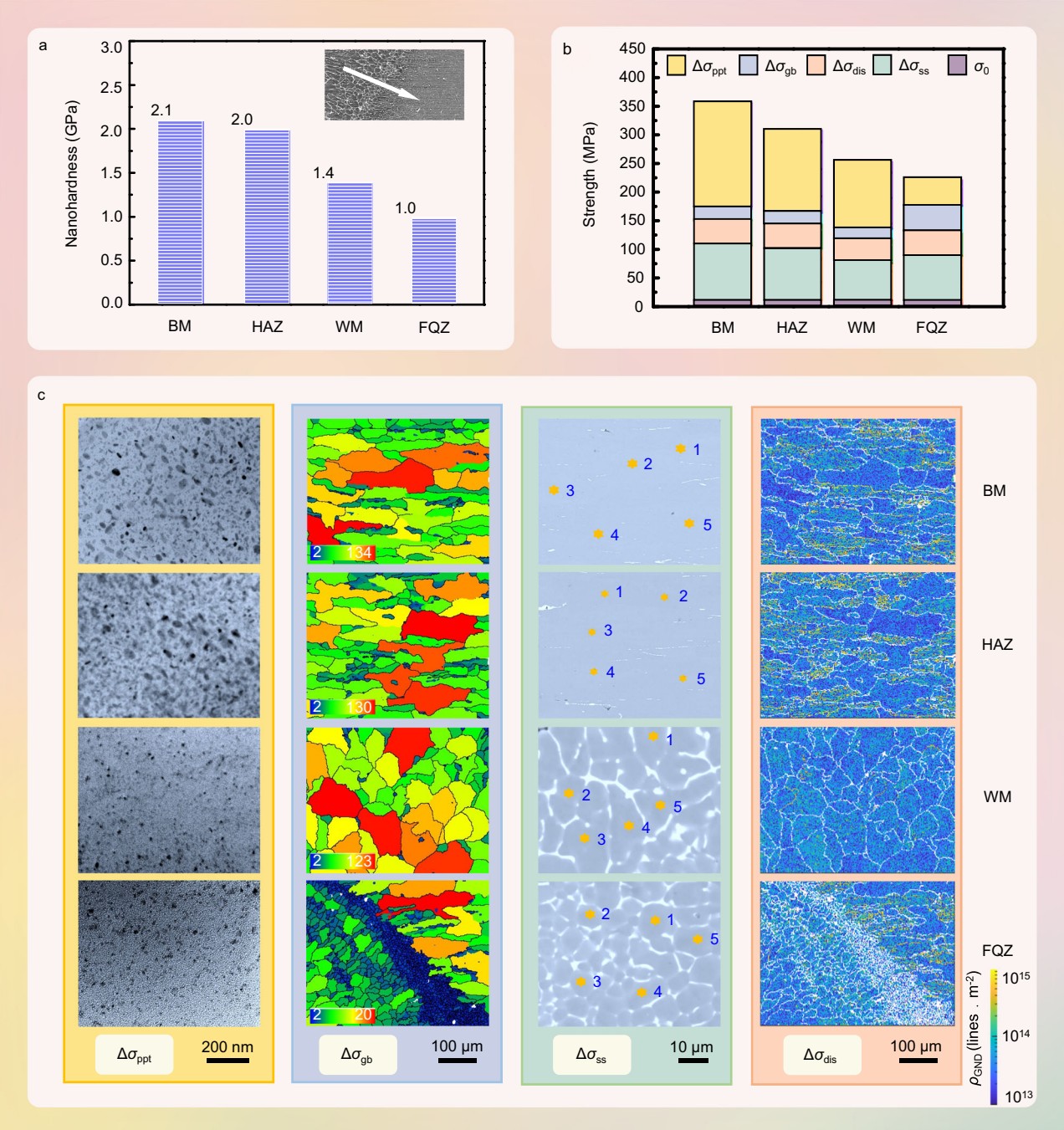

**Fig. 2 | Mechanical properties and microstructural features. a** The average nanoindentation hardness recorded for each region across the weld where the BM, HAZ, WM and FQZ represent base metal, heat affected zone, weld metal and fine equiaxed zone, respectively. **b** The estimated strengthening contributions arising from the precipitate ($\Delta\sigma_{ppt}$), grain size ($\Delta\sigma_{gb}$), solute ($\Delta\sigma_{ss}$) and dislocation ($\Delta\sigma_{dis}$) strengthening, respectively. $\sigma_0$ represents the baseline strength of pure aluminum. **c** (From left to right) distributions of precipitates in the grain interiors, distributions of grain sizes, scanning points used for chemical analysis by electron probe micro analysis (EPMA) and density of geometrically necessary dislocations ($\rho_{GND}$).

Hall-Petch, Fleischer, Bailey-Hirsch, and Orowan models, respectively (see Methods).

Conventionally, arithmetic addition (Eq. 1) and quadratic addition (Eq. 2) have most commonly been used to combine these strengthening contributions[13,14]:

$$\sigma_y = \sigma_0 + \Delta\sigma_{ss} + \Delta\sigma_{ppt} + \Delta\sigma_{gb} + \Delta\sigma_{dis} \quad (1)$$

$$\sigma_y = \sigma_0 + \Delta\sigma_{ss} + \sqrt{\Delta\sigma_{ppt}^2 + \Delta\sigma_{gb}^2 + \Delta\sigma_{dis}^2} \quad (2)$$

where $\sigma_0$ represents the baseline strength of pure aluminum ($\sigma_0 = \sim 10$ MPa[15]). We have considered both models to predict the yield strength of the BM giving strengths of ~450 MPa and ~430 MPa respectively.

Given the former is simpler, closer to the experimental value (~451 MPa in Supplementary Fig. 1) and because the main focus is on evaluating the relative importance of the individual mechanisms rather

**Table 1 | Values of the microstructural features critical to material strength for the four regions of the weld**

| Materials | Grain size | Solute concentrations | | | Dislocation densities | Precipitation size and volume fraction | |
|---|---|---|---|---|---|---|---|
| | $d$ (μm) | $c_{Zn}$ (wt %) | $c_{Mg}$ (wt %) | $c_{Cu}$ (wt %) | $\rho_{GND}$ (m$^{-2}$) | $r$ (nm) | $f$ (%) |
| BM | 37 | 6.27 | 2.18 | 2.55 | $8.82 \times 10^{13}$ | 15 | 3.7 |
| HAZ | 43 | 6.22 | 2.16 | 2.27 | $9.00 \times 10^{13}$ | 18 | 3.1 |
| WM | 45 | 2.83 | 1.76 | 0.66 | $8.45 \times 10^{13}$ | 12 | 1.3 |
| FQZ | 7 | 5.04 | 2.36 | 1.68 | $8.07 \times 10^{13}$ | 11 | 0.2 |

than finding a model that precisely matches the yield strength, we have used arithmetic addition to understand the relative contributions in each region of the weld. It has been used in many investigations[16–18]. The relative contributions are shown in Fig. 2b and in Supplementary Table 2.

Irrespective of the addition rule, it is evident that precipitation strengthening is the most significant contributor to the strength in the BM while grain size strengthening is relatively insignificant. Although the grain refinement in the FQZ means that the grain boundary strengthening is ~2 times larger, the loss in strength arising from the reduction in the precipitation strengthening (a quarter that for the BM) means that overall it is significantly softer.

It has been suggested that the evaporative loss of Zn and the inverse segregation of Cu are the main reasons for the low level of strengthening precipitates in the WM[19]. For the FQZ, the peak temperature is significantly lower than the peak temperature in the WM, so that significant evaporative loss of Zn is unlikely. Furthermore, the laminar boundary layer in the FQZ may suppress the inverse segregation of Cu[20]. As a result, the number of precipitates within the grains in the FQZ, which is much smaller than that for the WM (Fig. 2c), most probably arises from a combination of the extensive segregation of the strengthening elements to the very many grain boundaries (Fig. 1e) and the fast solidification together which limit the extent of reprecipitation of the precipitates on cooling.

## Damage evolution mechanism

The sharp variation in mechanical properties at the FQZ would be expected to lead to inhomogeneous plastic deformation and a high cracking sensitivity. An image-based 3D finite element (FE) simulation undertaken in ABAQUS confirms that, even at low stresses, plastic strain concentrates in the lower parts of the FQZ and the WM due to their low strengths (Fig. 3a). This explains why a crack is observed by time-lapse microCT (Fig. 3b) to initiate from the weld toe and then propagate approximately along the FB with increased loading (Supplementary Movie and Supplementary Fig. 2). Post-mortem fractography (Fig. 3c) reveals the fracture surface to be relatively flat exhibiting smooth curving facets. The size of these curved facets is consistent with the equiaxed grains inside the FQZ. This suggests that the crack grows primarily along the grain boounaries in the FQZ (Fig. 3c and Supplementary Fig. 3).

In order to visualize the 3D nature of the intergranular phases, a micropillar was excised from a region of FQZ by plasma FIB for an unstrained sample for examination by synchrotron X-ray nanoCT (Fig. 1c). The intergranular phases (rendered yellow in Supplementary Fig. 4 and Fig. 3d) include rod-shaped, network-shaped and dendritic morphologies. The different projections show the morphology of the intergranular phases to be anisotropic (Fig. 3e); presenting a network-like morphology on both the x-y and z-x planes, and a flake shape on the y-z plane, in which the y-z plane is parallel to the plane consisting of the weld thickness and welding direction. This indicates that the intergranular phases tend to occupy the spaces between the grains along the weld thickness.

In order to investigate the damage evolution micro-mechanism in greater detail, two micropillars (40 μm in diameter) were excised by plasma focused ion beam (PFIB) from the weld when $\sigma = 270$ MPa, as indicated in Fig. 3b, to perform synchrotron X-ray nanoCT observations (Fig. 3f and Supplementary Fig. 4). These show a large interconnected void (colored green) in the upper part and many isolated micro-voids (colored red) in the bottom half of the two micropillars, both types having initiated at the intergranular phase. The micro-voids are randomly distributed irregular, ellipsoidal or spherical shapes. Quantitative analysis shows that the equivalent diameter of these micro-voids is 100-500 nm, with an average value of ~240 nm, with those below 300 nm representing 90% of the distribution. Their projected area on the z-x plane is generally smaller than that on the x-y and y-z planes (Fig. 3e) suggesting they nucleate and grow along the welding direction perpendicular to the loading direction.

It is evident from Fig. 3g that all the micro-voids (red) nucleate from the intergranular phases (yellow) due to the plastic incompatibility. They form either within the intergranular phases (indicated by A) by phase fracture, or adjacent to it (indicated by B) suggesting debonding of the particle-grain boundary interface, or, most commonly, at the grain boundary triple junctions (indicated by C and D).

To identify the composition of the intergranular phases a typical interphase region was excised by PFIB for TEM analysis. A number of intergranular phase regions were located in bright field imaging and identified through a combination of SAED and EDS analysis. Representative results are shown in Fig. 3h and Supplementary Fig. 5 which are consistent with AlCuMg. This is not unexpected, brittle AlCuMg phases are often observed in Al alloys with higher Cu content, such as the 2000-series Al-Cu alloy. Here we have very high Cu concentrations due to the segregation and this explains the presence of AlCuMg phases. The plastic mismatch between these hard regions and the soft grain interiors promotes void nucleation at the grain boundaries (Fig. 3 and Supplementary Fig. 6).

Unsurprisingly, the high resolution EBSD based geometrically necessary dislocation (GND) density maps in Supplementary Fig. 7 show that the GND density in the FQZ has increased significantly when the applied stress reaches 320 MPa (slightly above the yield strength of the FQZ in Supplementary Table 2). It is noteworthy that the increased GND density is not concentrated at the grain boundaries which serve as strong barriers for dislocation movement, but rather they are relatively homogeneously distributed in the grain interiors and at the grain boundaries. This is also confirmed by EBSD measurements recorded in situ during tensile straining (see Supplementary Fig. 8).

## A hybrid welding technique

Clearly the FQZ poses a significant threat to the reliable in-service performance of welded structures, however, research indicates that it cannot be entirely eliminated by varying the welding parameters[6]. A key question therefore is, how can we re-engineer the FQZ to mitigate its deleterious effect on weld performance? The first strategy to consider is post-weld heat treatment (PWHT). An in situ heating EBSD experiment shows that the PWHT (470 °C/30 min) had essentially no effect on the FQZ, for the surface grains at least (Supplementary Fig. 9). This observation is supported by destructive observations of the FQZ

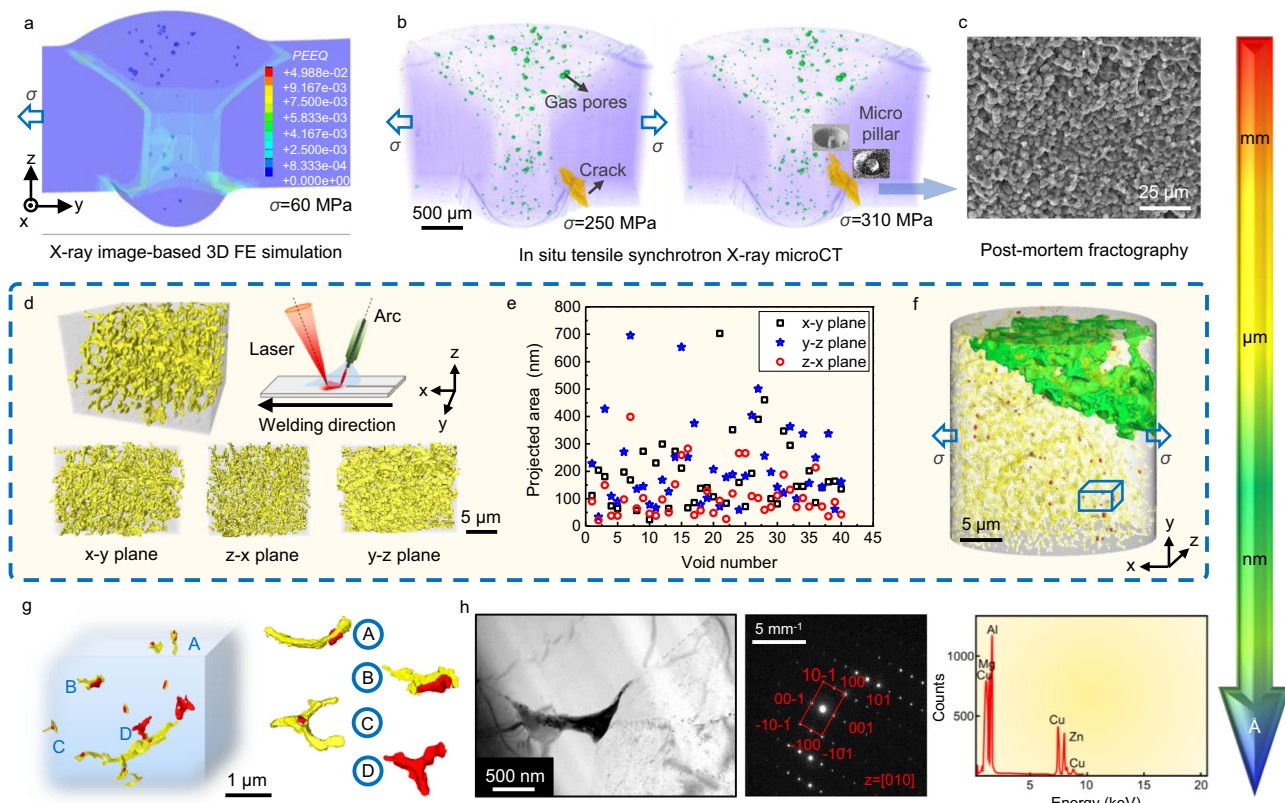

**Fig. 3 | Multiscale correlative tomography workflow applied to detect damage nucleation and evolution inside the fine equiaxed zone (FQZ). a** Distribution of equivalent plastic strain (PEEQ) across the weld zone predicted by an image-based 3D finite element (FE) simulation where the applied stress $\sigma = 60$ MPa perpendicular to the weld. **b** 3D volume renderings (pores rendered green and cracks yellow) acquired at two loading stages by in situ tensile synchrotron micro computed tomography (microCT) to determine the preferential damage nucleation site and damage evolution. **c** Post-mortem fractography of a failed welded joint by scanning electron microscope (SEM). **d** NanoCT volume renderings of the spatial distribution of intergranular phases prior to deformation. **e** Quantitative analysis of the projected areas of nucleated micro-voids when $\sigma = 270$ MPa. **f** Spatial distribution of intergranular phases and nucleated micro-voids by high-resolution synchrotron X-ray nanoCT when $\sigma = 270$ MPa showing the large-sized long-range connected voids (green), the intergranular phases (yellow) and the nucleated micro-voids (red). **g** NanoCT visualization of the interaction between the intergranular phases (yellow) and nucleated micro-voids (red). **h** Intergranular phase inducing micro-void nucleation observed by (transmission electron microscopy-energy dispersive spectrometry) TEM-EDS, showing TEM image, selected area electron diffraction (SAED) and EDS spectra.

and the fact that after PWHT the yield behavior (ultimate tensile and yield strengths of $\sigma_b = 375 \pm 6$ MPa and $\sigma_{p0.2} = 295 \pm 5$ MPa) was similar to that ($\sigma_b = 406 \pm 11$ MPa and $\sigma_{p0.2} = 292 \pm 10$ MPa) of the as-welded joints.

An alternative strategy is to enhance molten pool turbulence so as to modify the weld microstructure[21,22]. Here we investigated the effect of oscillating the laser beam during HLAW and applying a pulsed magnetic field (OLHW + m) immediately following welding after the solid has formed. In this way we have produced butt-welded joints having the wavy morphology shown in Fig. 4a. At the microscale, the EBSD IPF maps in Fig. 4d show that the FQZ at the fusion boundary has been broken up. This is evident in Fig. 4d where it is alternately arranged with coarse equiaxed dendritic structures on the *z-x* section and intermittently distributed near the FB on the *y-z* section. Its curved nature can also be seen in the *x-y* section.

The tensile strength of the OSHW + m weld (~470 MPa) is clearly significantly (20%) better than the conventional HLAW (Fig. 4b) and unsurprisingly, far superior (60% higher) than that made by gas metal arc welding (GMAW). For comparison it is slightly greater than for welds made by friction stir welding (FSW) ($\sigma_b = 450$ MPa[23]) after the same natural aging treatment (~3000 h). The fracture section in Fig. 4d (right) shows that the crack propagates initially the soft FQZ but then grows into the weld zone despite its higher strength. This is because in this region the primary crack has to deflect to continue to propagate due to the curved and discontinuous nature of the FQZ. As a result, the

gradient structure (mixture of FQZ and WM) near the fusion boundary for the OSHW + m weld displays a higher cracking resistance compared to conventional HLAW having a straight FQZ[24].

Given that the loss of precipitate strengthening in the FQZ is critical to the extensive softening of the FQZ it is important to compare the precipitate microstructures for the modified and conventional hybrid laser welds (Supplementary Fig. 10). It is evident that the number of precipitates in the FQZ is much higher than for the conventional weld, with volume fractions of ~0.9% and ~0.2%, respectively. This is because the laser beam oscillation more effectively controls the distribution of alloying elements in the weld and mitigates macro-segregation[25]. This retains the elements in solution leading to their subsequent reprecipitation as strengthening precipitates in the interior of the grains and consequently higher strength.

As a result of the FQZ morphology, the crack follows a more tortuous macroscopic failure path in comparison with the HLAW joints leading to a relatively slower crack growth rate Fig. 4c, especially in the microstructure sensitive near-threshold and unstable crack growth regions. The crack deflection gives rise to several periods of arrested growth, probably due to the crack growing through a mixture of a harder phase (HAZ and WM) and a softer phase (FQZ) regions near the fusion boundary[26,27].

In summary, we have quantified the strengthening mechanisms giving rise to significant softening in the FQZ and delineated the sequence by which the crack initiates intergranularly due to the

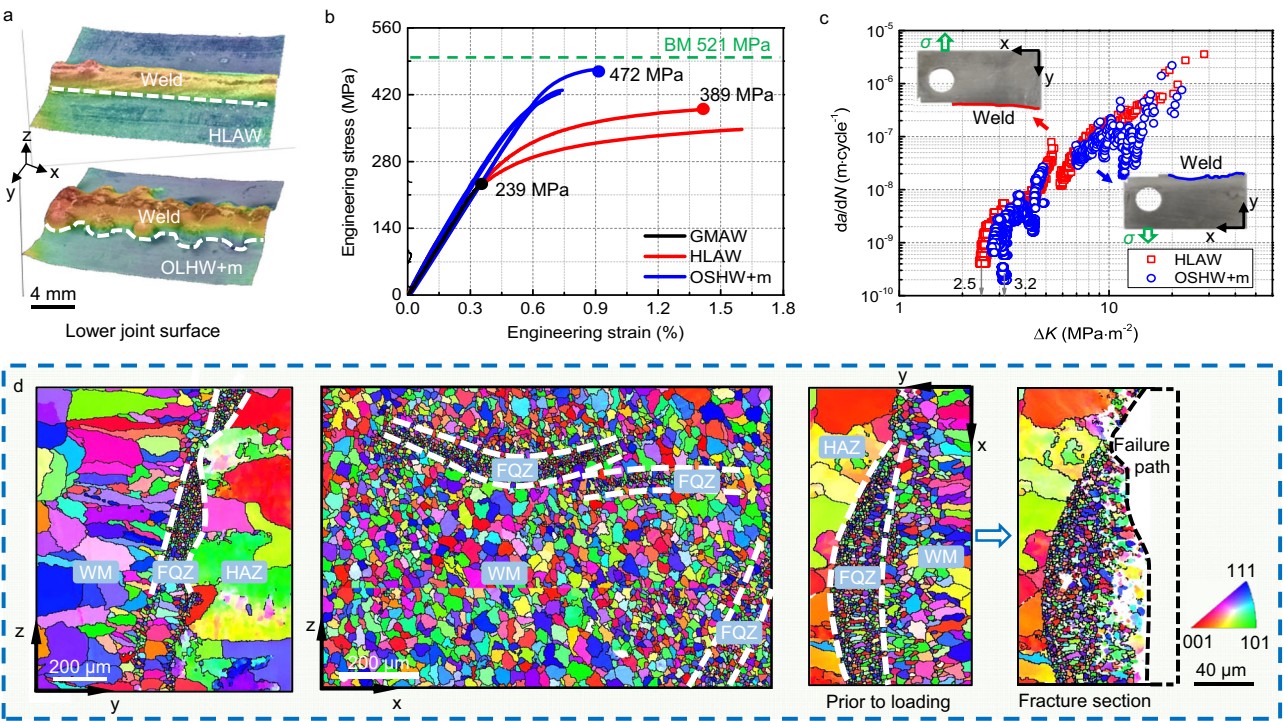

**Fig. 4 | Microstructural features and mechanical properties of the welded joints manufactured by the traditional hybrid laser and arc butt weld (HLAW) and the proposed oscillating laser hybrid welding with pulse magnetic field (OSHW + m). a** Morphology of the lower weld surface using ultra-depth 3D microscopy. **b** Engineering stress-strain curves of traditional gas metal arc weld (GMAW) joints, HLAW joints and OSHW + m joints. **c** Fatigue crack growth rate curves ($da/dN$-$\Delta K$ where $da/dN$ is the fatigue crack growth rate, $a$ is the crack length, $N$ is the number of loading cycles, $\Delta K$ is the stress intensity factor range) of HLAW joints and OSHW + m joints with side views of the macro failure path. The applied stress $\sigma$ is perpendicular to the weld. **d** (From left to right) Electron backscatter diffraction (EBSD) X inverse-pole figure (IPF) maps on y-z, z-x (where the weld root is located) and x-y sections of OSHW + m joints. The WM, FQZ and HAZ represent weld metal, fine equiaxed zone and heat affected zone.

presence of AlCuMg phases. To counter the deleterious effect of the FQZ on the low strength of hybrid laser welded 7050 aluminum alloys, we have introduced an oscillated laser hybrid weld with externally applied magnetic field. This process disrupts the FQZ distribution in three dimensions and increases the precipitate density. Together these changes radically increase the tensile strength extending the UTS to ~470 MPa. This is ~90% of that of the base metal (UTS = 521 MPa), comparing favorably with the strength of the solid state friction stir welded joints[28,29].

## Methods
### Materials and welding procedure
A hybrid fiber laser-pulsed arc welding system was employed to produce 2-mm-thick 7050 Al alloy butt-welded joints, where the weld was perpendicular to the rolling direction. The filler material was ER5356 Al-Mg wire having a diameter of 1.2 mm. The welding parameters were: laser power $P = 3$ kW, electric current $I = 100$ A, welding speed $\upsilon = 6$ m/min and defocusing distance $\Delta = -1$ mm.

A second HLAW experiment (designated OSHW + m) was undertaken but employing laser beam oscillation following a saw-tooth trajectory down the weld. The optimized welding parameters were: laser power, 6 kW; welding speed, 8 m/min; defocusing distance, +2 mm; oscillating diameter, 1-3 mm; and oscillating frequency, 100-300 Hz. Post-weld electropulsing treatment was then applied to the OSHW joints immediately following welding. Electropulsing was achieved by passing and electrical current through a material using a home-made device equipped with a capacitor bank discharge circuit (Fig. 1a). We chose a discharge voltage of 4 V, an alternating current pulse with duration 100 ms, followed by a 10 s natural air cooling.

### Microstructural characterization
For EBSD analysis, the samples were polished with colloidal silica and milled by Ar ion beam. The crystallographic data was estimated using a TESCAN MIRA3 SEM equipped with a Bruker e-Flash FS detector. In order to calculate the GND density, cross-correlation based EBSD method was adopted. In the current work, 12-bit 1200 × 1200 pixel patterns were saved to hard disk during EBSD scan. The calculated rotation gradients could be linked to Nye's dislocation tensor[30] and the dislocation density was estimated using L1 optimization by minimizing the total dislocation line energy. A full description of the method used for GND calculation can be found in Supplementary material[31].

The samples for EBSD analysis were further analyzed to determine the variation in alloying constituents near the fusion boundary and the elemental distributions across the grains and the grain boundaries using a JEOL JXA-8230 EPMA. The content of the solute elements in the interior of the grains was also quantitatively analyzed. Both the size and volume fraction of the precipitates within the grains were examined by a Tecnai G2 F30 S-TWIN TEM on the 20-μm-thick samples prepared by an Ion Beam Thinner. The chemical maps were acquired by EDS using four Bruker SDD detectors.

### Mechanical testing
A G200 KEYSIGHT Nano Indenter tester was used to perform nanoindentation hardness testing. The sample surface was carefully polished to achieve a surface roughness a quarter of the maximum indentation depth[32]. The maximum indentation depth was 1000 nm. The Oliver-Pharr method was adopted to calculate the nanohardness of different zones of the joints[33].

Monotonic tensile testing was conducted on three kinds of welded joints manufactured by GMAW, HLAW and OSHW + m, with a minimum width of 12 mm, a gage length of 65 mm and a thickness of 2 mm. The nominal strain rate was 1.0 mm/min. The loading direction was perpendicular to the welding direction. An extensometer with a gauge length of 20 mm was used to determine the yield stress and elastic modulus.

Fatigue crack growth rate testing was conducted on compact tension (CT) specimens cut from both HLAW and OSHW + m joints, using a frequency of 10 Hz and a load ratio of 0.1. The thickness and width of the CT specimens were 2 mm and 48 mm, respectively. A 10-mm long pre-crack was emanated from the notch tip along the weld. The crack growth rate (d$a$/d$N$) was determined by the 7-point increment polynomial method.

## Multiscale correlative tomography

We have threaded together in situ X-ray micro tomography, X-ray nano tomography and TEM-EDS characterization all for the same region. In situ tensile SR-μCT was performed on a specimen with an area of minimum section of 2 mm$^2$. The role of the large volume in situ tensile SR-μCT was to provide a means for identifying and locating RoI containing interesting features for higher resolution X-ray nano-tomography. A FEI Helios PFIB system[34] based on Xe$^+$ was used to extract a pillar of approximately 40 μm diameter and 40 μm height, at the location of the RoI, for nanoCT. To characterize the spatial distribution of the nucleated micro-voids and intergranular phases, a nanoscale X-ray CT experiment was performed on the BL4W1A at the Beijing Synchrotron Radiation Facility (BSRF), China. The 'large field of view' mode was adopted with a field-of-view size of 60 μm × 60 μm, a pixel size of 64.1 nm, a resolution of 100 nm, an energy of 8 keV and an exposure time of 12 s. All visualization was performed using the Avizo software package. To further determine the critical intergranular phases inducing the micro-void nucleation, thin slices of the RoI were removed for analysis in the TEM using a Tecnai G2 F30 S-TWIN TEM.

## X-ray CT Image-based finite element modeling

The volumetric image datasets of metallurgical defects from SR-μCT were segmented using the tomography software Avizo. Considering the meshing process and computing feasibility, the sharp variations of defects were smoothed by adjusting the smooth factor. Avizo-based 3D reconstruction was able to directly produce surface meshes (linear triangles) of inner defects and outer surfaces of the samples. Then the volume filling of the samples was conducted using the software HyperMesh, producing over 2,657,544 linear tetrahedral elements (C3D4) for FE analysis. Both linear elastic and elasto-plastic responses were considered, and the Young's modulus and Poisson's ratio were 70 GPa and 0.33, respectively. The bilinear isotropic model was employed for the elastic-plastic prediction where the stress–strain curves were determined by the load-displacement curves recorded during nanoindentation testing using the reverse analysis based on dimensionless functions. The right surface in Fig. 3a was fixed in all directions and a remote stress of 60 MPa in $y$ direction was applied on the left surface.

## Crystal plasticity finite element modelling

The modeling was performed on a model with a dimension of 6.4 μm × 4.29 μm × 0.19 μm, consisting of eight grains and 64,437 C3D8R elements using ABAQUS/CAE. The grain orientations were assigned in terms of three Euler angles, {θ, φ, Ω}, representing rotations from the crystal basis to the global basis, according to the EBSD results. The morphology of intergranular phases was referred to the bright-field TEM images. The boundary conditions are: on the $x = 0$ surface, the displacement in the $x$ direction is zero ($u_x = 0$); on the $y = 0$ surface, $u_y = 0$; on the $z = 0$ surface, $u_z = 0$; on the $y = 6.4$ μm and $x = 0.19$ μm surfaces, the traction is zero; on the $z = 4.29$ μm surface,

the velocity in the $z$ direction is constant, corresponding to a strain rate of $4 \times 10^{-5}\,\text{s}^{-1}$. Each intergranular phase is regarded as a rigid body through the rigid body constraint. The face-centered cubic structure of α-Al matrix is anisotropic and defined by three elastic constants $C_{11}$, $C_{12}$, and $C_{44}$, where $C_{11} = 106.43$ GPa, $C_{12} = 60.35$ GPa, and $C_{44} = 28.21$ GPa[35].

The nonlocal crystal plastic constitutive laws in the DAMASK simulation platform developed by the Max-Planck-Institut für Eisenforschung in Germany were adopted[36]. The modelling is based on the dislocation mechanism. The dislocation flow term is derived from the relationship between the plastic strain gradient and the geometrically necessary dislocation density. The dislocation density evolution includes dislocation generation, dislocation annihilation, dislocation pair formation and annihilation, and dislocation flow between material points. The plastic slip rate is described by the Orowan equation:

$$\dot{\gamma}^\alpha = \sum \rho^\alpha_{\text{mobile}} b v^\alpha \tag{3}$$

where $\rho^\alpha_{\text{mobile}}$ is mobile dislocation density, $v^\alpha$ is the dislocation velocity and the value of the Burgers vector $b$ is $2.48 \times 10^{-10}$ m.

Furthermore, the dislocation multiplication can be expressed as:

$$_{mult}\dot{\rho}^\alpha = \frac{k_1(|\dot{\gamma}^\alpha_{e+}| + |\dot{\gamma}^\alpha_{e-}|) + (|\dot{\gamma}^\alpha_{s+}| + |\dot{\gamma}^\alpha_{s-}|)}{bk_2\lambda^\alpha} \tag{4}$$

where $\dot{\gamma}^\alpha_{e+}$, $\dot{\gamma}^\alpha_{e-}$, $\dot{\gamma}^\alpha_{s+}$ and $\dot{\gamma}^\alpha_{s-}$ are the plastic shear rate for different types of dislocation. $k_1$ and $k_2$ are the proportion coefficient. $\lambda^\alpha$ is the mean free path of dislocation. The contribution coefficient, $k_1$, and multiplication coefficient, $k_2$, are $k_1 = 0.1$ and $k_2 = 10$.

However, the movement of dislocation on the slip plane is hindered by the forest dislocation. The effective shear stress on the slip plane is:

$$\tau_{eff} = |\tau^\alpha| - \tau_{cr} \tag{5}$$

where $\tau^\alpha$ is the resolved shear stress along the slip plane of the second Piola-Kirchhoff stress.

In addition, the interaction strength between dislocations is:

$$\tau_{cr} = Gb\sqrt{\sum_{\alpha=1}^{N_{slip}} \xi^{\alpha\alpha'} \rho^\alpha} \tag{6}$$

where $\xi^{\alpha\alpha'}$ is the dislocation interaction coefficient among slip systems. $G$ is the shear modulus.

The relationship between the plastic distortion tensor $\boldsymbol{\beta}^\alpha$ and the dislocation density is associated by the Nye tensor $\boldsymbol{\alpha}^\alpha$:

$$\sum \rho^\alpha l^\alpha \otimes b = \boldsymbol{\alpha}^\alpha = -curl\boldsymbol{\beta}^\alpha \tag{7}$$

where $l^\alpha$ represents the unit vector along the dislocation line on the α-slip system.

By defining the dislocation flux $f^\alpha$ as the product of the dislocation density and the dislocation velocity, the flow relationship of dislocations between material points can be derived:

$$\partial_t \rho^\alpha + div f^\alpha = 0 \tag{8}$$

## Estimating the strengthening contributions

The yield strength increase arising from the Orowan by-passing mechanism can be expressed as follows[37]:

$$\Delta\sigma_{\mathrm{ppt}} = \frac{0.4MGb}{\pi\sqrt{1-\upsilon}}\frac{\ln(2\bar{r}/b)}{L_{\mathrm{p}}} \tag{9}$$

where $M = 3.06$ is the Taylor factor of face-centered-cubic (fcc) alloy. $G = 27$ GPa is the shear modulus of aluminum alloy. $b = 0.286$ nm is the value of the Burgers vector of Al alloy. $\upsilon = 0.33$ is the Poisson's ratio of aluminum alloy. $\bar{r} = \sqrt{2/3}r$ in which $r$ is the mean precipitates radius. $L_{\mathrm{p}} = 2\bar{r}(\sqrt{\pi/4f} - 1)$ is the inter-precipitate distance in which $f$ is the precipitates volume fraction. In our case, both $r$ and $f$ are extracted from the TEM images using image processing software ImageJ.

The strengthening contribution arising from the grain boundaries can be calculated using the classical Hall-Petch equation[38]:

$$\Delta\sigma_{\mathrm{gb}} = k \cdot d^{-1/2} \tag{10}$$

where $k$ is the strengthening coefficient specific to each material and $k = 0.12$ for aluminum alloy. $d$ is the grain size derived from on the EBSD analysis.

The strengthening contribution arising from solute strengthening has been estimated using the Fleischer equation[39]:

$$\Delta\sigma_{\mathrm{ss}} = \sum_i \Delta\sigma_i c_i \tag{11}$$

where $\Delta\sigma_i$ is the theoretical strengthening efficiency for each element ($\Delta\sigma_{\mathrm{Zn}} = 2.9$ MPa/wt. %, $\Delta\sigma_{\mathrm{Mg}} = 18.6$ MPa/wt. % and $\Delta\sigma_{\mathrm{Cu}} = 13.8$ MPa/wt. %[39]). $c_i$ is the concentration of the solute element (in wt. %), which is measured by EPMA in the study.

The Bailey-Hirsch equation[40] is adopted to evaluate the dislocation strengthening due to the interaction of dislocations:

$$\Delta\sigma_{\mathrm{dis}} = M\alpha \, Gb\rho_{\mathrm{GND}}^{1/2} \tag{12}$$

where $\alpha$ is a material constant ($\alpha = 0.2$ of the fcc alloy). $\rho_{\mathrm{GND}}$ is the GND density, which is determined by EBSD analysis.

## Data availability

The experimental data that support the findings of this study are available from the corresponding authors upon request. The source data underlying Figs. 1c, 2c, 3a, b, d–f, 4b, c are available in the database under accession code 0101 [https://pan.baidu.com/s/12nfseoV38kelFSYTZ_8k5Q].

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

## Acknowledgements

The authors sincerely thank the National Natural Science Foundation of China (U2032121 and 12192212). As an honorary staff, SC Wu is grateful for his senior visiting position at the Henry Royce Institute for Advanced Materials funded by EPSRC (EP/R00661X). AM Korsunsky thanks the ESPRC support (EP/V007785/1). PJ Withers also acknowledges a European Research Council Advanced Grant (CORREL-CT, Grant No 695638).

## Author contributions

Y.N.H., S.C.W., Y.K.Y., T.Q.X., and Q.X.Y. conducted in situ SR-μCT experiments and analyzed the resultant data. Y.N.H., Y.G., S.C.W., Z.S., A.M.K., X.L.Z. and P.J.W. characterized microstructures. Y.N.H., S.C.W., and Y.N.F. performed mechanical tests. X.Z. performed the modeling. Y.N.H., S.C.W. and Z.G.C. were involved with processing development and sample weld. Y.N.H., S.C.W., Z.S., S.L.-P., X.Q.Z., and P.J.W. drafted the initial manuscript. S.C.W., G.Z.K., and P.J.W. conceived, designed, and led the project. All co-authors contributed to the data analyses.

## Competing interests

The authors declare no competing interests.
