## [Peer Review File · Nature Communications]

Title: Inhibiting weld cracking in high-strength aluminium alloysREVIEWER COMMENTS

Reviewer #1 (Remarks to the Author):

It is a very good paper with comprehensive results and discussion of the work carried out on the system.

I have only 2 comments on the paper:

A "Movie S1" is referred to in the paper but no movie file was supplied. Perhaps remove reference if not included for readers to review.

There are issues with the pinning effect of a free surface on grain boundaries which is a particular issue with Al and its alloys. The conclusions drawn from Fig. S9 therefore may not be representative of what is occurring inside a bulk sample.

Reviewer #2 (Remarks to the Author):

The manuscript proposed a novel hybrid welding technique-laser beam oscillation with a pulsed magnetic field to inhibit the cracking from a fine equiaxed zone in the 7xxx series aluminum alloy welded joints. The mechanism of softening in the FQZ was fully studied using several advanced analysis methods including EMPA, HR-EBSD, SR- μ CT, nano-CT, HR-TEM. The manuscript was well-structured and written. However, the following points should be addressed before publication:

1. In the introduction, only FQZ was mentioned. Generally, the softening problems of 7xxx Al alloys is also related to the evolution of precipitates. According to the results of the manuscript, the influence of precipitates was surely significant. Please introduce the precipitates in the introduction.

2. "It is somewhat greater than those made by friction stir welding (FSW) (UTS = 450 MPa) under exactly the same heat treatment.", was the energy input from the plus magnetic field considered?

3. The oscillating laser-plus magnetic field broke the continuous FQZ, which had positive effects on the joint strength. However, as stated by the authors, "Although the grain refinement in the FQZ means that the grain boundary strengthening is ~ 2 times larger, the loss in strength arising from the reduction in the precipitation strengthening (a quarter that for the BM) means that it is significantly softer." Is there any influence of laser-plus magnetic field on precipitates evolution ?

4. The author should confirm the reasonability of "Assuming that these mechanisms contribute additively". A reference can be considered.

5. The brittle AlCuMg phases are the key factors leading to void nucleation and intergranular failure. The

reproducibility for determining the phases inducing void nucleation should be detailed.

6. A novel hybrid fabrication strategy is proposed exploiting laser beam oscillation and a pulsed magnetic field. This achieves a wavy and interrupted FQZ which contributes to a higher crack growth resistance or ductility. While it is also necessary to clarify the reasons for the improvement in strength. Besides, only one stress-strain curve is provided for each material. The reproducibility should be detailed.

7. The damage accumulation sequence has been investigated by time-lapse microCT. Are there any 3D visualization pictures or videos showing the damage sequence?

8. Change KAM maps in Fig. 2c to GND density maps since the softening behavior depends on dislocation density. The crystal plasticity finite element simulation results are shown in Fig. S6. However, the detailed modelling is not provided, the constitutive model in particular.

Detailed Response to Reviewers' Comments

“Inhibiting weld cracking in high-strength aluminium alloys”

We very much appreciate the reviewer's comments and suggestions (listed in blue type below), which have significantly helped us improve our paper. We have made substantial revisions to the original manuscript with all these suggestions incorporated. The detailed responses and the revisions made are given in black type below. The resulting changes are highlighted in yellow in the revised manuscript.

Best regards, and thanks,

Professor Philip J. Withers FRS, FREng., FIMMM, CAE Foreign Member

Chief Scientist and Regius Professor of Materials | Henry Royce Institute

Director of the National Research Facility for Lab. CT (NXCT.ac.uk)

E:p.j.withers@manchester.ac.uk T:+44 (0)161 306 4282

A: Royce Hub Building, The University of Manchester, Manchester, M13 9PL

Twitter: @RoyceInstitute LinkedIn: Henry Royce Institute

To Reviewer #1

General Comments:

It is a very good paper with comprehensive results and discussion of the work carried out on the system. I have only 2 comments on the paper.

Specific Comments:

1. A "Movie S1" is referred to in the paper but no movie file was supplied. Perhaps remove reference if not included for readers to review.

Action: *We have provided an animation (Supplementary Movie) showing damage development from crack nucleation through growth to final failure in the supplementary information.*

2. There are issues with the pinning effect of a free surface on grain boundaries which is a particular issue with Al and its alloys. The conclusions drawn from Fig. S9 therefore may not be representative of what is occurring inside a bulk sample.

This is a good point, the EBSD of itself does not provide conclusive proof that the FQZ

is little affected by heat treatment because of the potential for surface grain boundary pinning. However cross sections showed similar results and the monotonic tensile test after PWHT is similar to that before PWHT corroborating that PWHT does not ameliorate the effect of the FQZ.

Action: *A comment to this effect has been added to the paper.*

Fig. 1. Comparison of stress strain response of welds prior, and subsequent to, PWHT.

To Reviewer #2

General Comments:

The manuscript proposed a novel hybrid welding technique-laser beam oscillation with a pulsed magnetic field to inhibit the cracking from a fine equiaxed zone in the 7xxx series aluminum alloy welded joints. The mechanism of softening in the FQZ was fully studied using several advanced analysis methods including EMPA, HR-EBSD, SR- μ CT nano-CT, HR-TEM. The manuscript was well-structured and written. However, the following points should be addressed before publication.

Specific Comments:

1. In the introduction, only FQZ was mentioned. Generally, the softening problems of 7xxx Al alloys is also related to the evolution of precipitates. According to the results of the manuscript, the influence of precipitates was surely significant. Please introduce the precipitates in the introduction.

This is a very good point.

Action: *As you suggested, we have added a discussion of the effects of the evolution of the strengthening precipitates to the introduction.*

2. “It is somewhat greater than those made by friction stir welding (FSW) (UTS = 450 MPa) under exactly the same heat treatment.”, was the energy input from the plus magnetic field considered?

Because of the role of the strengthening precipitates, the post-weld aging can have a considerable effect on the ultimate tensile and yield strengths of welded joints. Therefore, both the friction stir welds and oscillating laser hybrid welds with pulse magnetic field were given exactly the same post-weld natural aging treatment for comparison. The effect from the energy input from the plus magnetic field is not considered as this is relatively small compared to the energy provided by the lasers and the subsequent aging.

Action: *For comparison it is slightly greater than for welds made by friction stir welding (FSW) ($\sigma_b = 450$ MPa) after the same natural aging treatment (~3000 h).*

3. The oscillating laser-plus magnetic field broke the continuous FQZ, which had positive effects on the joint strength. However, as stated by the authors, “Although the grain refinement in the FQZ means that the grain boundary strengthening is ~2 times larger, the loss in strength arising from the reduction in the precipitation strengthening (a quarter that for the BM) means that it is significantly softer.” Is there any influence of laser-plus magnetic field on precipitates evolution?

Given that loss of precipitate strengthening is the biggest influence on strength this is a good question. To answer this we have undertaken TEM characterization to explore the precipitates in the FQZ, and the bright-field TEM images shown below alongside the distribution of precipitates in the FQZ of conventional hybrid laser-arc welds.

Fig. 2. Bright-field TEM images of precipitates in the interior of grains in the FQZ. a, oscillating laser hybrid welds with pulse magnetic field. b, conventional laser hybrid weld.

It can be observed that the number of precipitates in the FQZ of newly proposed welds is significantly higher than for the conventional weld, with the volume fractions of ~0.9%

and ~0.2%, respectively. In the conventional fusion welds, the FQZ is mainly located along the fusion boundaries where the cooling rate is very high. On the one hand, the high solidification rates enable severe segregation of Zn, Mg and Cu to the grain boundaries. On the other hand, the fast cooling inevitably limits the reprecipitation of the precipitates. Therefore, there is a significant reduction in the number of precipitates in the interior of the grains and consequently severe softening of FQZ.

Previous investigations have revealed that high-frequency beam oscillation could help to control the distribution of alloying elements in the weld [1] with macro-segregation found for laser-arc hybrid welds mitigated by using beam oscillation [2]. Improved solute transfer and reduced macro-segregation have been attributed to stirring and vortex suction effects produced by the high-frequency oscillating beam. The reduced segregation promotes solution of the elements and reprecipitation of the strengthening precipitates inducing more precipitates in the interior of the grains.

1. Gao, M., Zhang, Y. Z. & Meng, Y. F. Interface homogenization and its relationship with tensile properties of laser-arc hybrid welded Al/steel butt-joint via beam oscillation. *J. Mater. Sci.* **56**, 14126-14138 (2021).
2. Wang, L. *et al.* A pathway to mitigate macrosegregation of laser-arc hybrid Al-Si welds through beam oscillation. *Int. J. Heat Mass Tran.* **151**, 119467 (2020).

Action: *We have added side by side TEM micrographs (Fig. S10) to show the higher precipitate density in the oscillating laser hybrid welds with pulse magnetic field compared to the conventional hybrid laser weld and discussed the origins of this effect in the main text.*

4. The author should confirm the reasonability of “Assuming that these mechanisms contribute additively”. A reference can be considered.

The best way of combining the different strengthening mechanisms (grain size strengthening ($\Delta\sigma_{gb}$), solid-solution strengthening ($\Delta\sigma_{ss}$), dislocation strengthening, ($\Delta\sigma_{dis}$) and precipitation strengthening ($\Delta\sigma_{ppt}$)) has been the subject of much debate in the literature. Conventionally, arithmetic addition of Eqn. (1) and quadratic addition of Eqn. (2) have been the main contenders for mathematically describing strengthening mechanism superposition [3].

$$\sigma_y = \sigma_0 + \Delta\sigma_{ss} + \Delta\sigma_{ppt} + \Delta\sigma_{gb} + \Delta\sigma_{dis} \quad (1)$$

$$\sigma_y = \sigma_0 + \Delta\sigma_{ss} + \sqrt{\Delta\sigma_{ppt}^2 + \Delta\sigma_{gb}^2 + \Delta\sigma_{dis}^2} \quad (2)$$

We have considered both models to calculate the yield strength of the BM. The predicted yield strengths are ~451 MPa and ~432 MPa based on Eqns. (1) and (2),

respectively. Given the former is simpler, a little closer to the experimental value (~465 MPa) and because the main focus is on evaluating the relative importance of the individual mechanisms, we have used the arithmetic addition of Eqn. (1). The simple arithmetic addition is also generally used in some investigations [4-8].

3. Ferguson, J. B. *et al.* On the superposition of strengthening mechanisms in dispersion strengthened alloys and metal-matrix nanocomposites: Considerations of stress and energy. *Met. Mater. Int.* **20**, 375-388 (2014).
4. Fang, H. J. *et al.* Evolution of texture, microstructure, tensile strength and corrosion properties of annealed Al-Mg-Sc-Zr alloys. *Mater. Sci. Eng. A* **804**, 140682 (2021).
5. Lee, S. H. *et al.* Precipitation strengthening in naturally aged Al-Zn-Mg-Cu alloy. *Mater. Sci. Eng. A* **803**, 140719 (2021).
6. Wang, J. *et al.* Microstructure evolution and mechanical properties of the electron-beam welded joints of cast Al-Cu-Mg-Ag alloy. *Mater. Sci. Eng. A* **801**, 140363 (2021).
7. Masuda, T. *et al.* Achieving highly strengthened Al-Cu-Mg alloy by grain refinement and grain boundary segregation. *Mater. Sci. Eng. A* **793**, 139668 (2020).
8. Sunde, J. K. *et al.* Linking mechanical properties to precipitate microstructure in three Al-Mg-Si(-Cu) alloys. *Mater. Sci. Eng. A* **807**, 140862 (2021).

Action: *We have discussed the choice of strengthening model in more detail and added references as suggested by the referee.*

5. The brittle AlCuMg phases are the key factors leading to void nucleation and intergranular failure. The reproducibility for determining the phases inducing void nucleation should be detailed.

The multiscale approach we followed enabled us to observe the critical role of the brittle phases in void nucleation and intergranular failure. The nature of the intergranular phases was then investigated by diffraction and EDS analysis in the TEM on an extended region excised using a PFIB. Multiple sites were analyzed showing good reproducibility. Fig. S5 corroborates the results shown in Fig. 3h that the phase is AlCuMg.

Fig. 3. Intergranular phase inducing micro-void nucleation observed by TEM-EDS characterization, showing bright field TEM image, SAED and EDS spectra. a, TEM image and SAED. b, EDS spectra.

Action: *Comment on the reproducibility has been added to the main text.*

6. A novel hybrid fabrication strategy is proposed exploiting laser beam oscillation and a pulsed magnetic field. This achieves a wavy and interrupted FQZ which contributes to a higher crack growth resistance or ductility. While it is also necessary to clarify the reasons for the improvement in strength. Besides, only one stress-strain curve is provided for each material. The reproducibility should be detailed.

The improved strength is closely related to the evolution of precipitates and by interrupted the crack path achieved by breaking up the FQZ. The changes to the precipitate distribution are shown in Fig. 2 and explained in response to question 3.

Action: *Multiple engineering stress-strain curves for the OSHW+m welds are now shown in Fig. 4b in the revised manuscript.*

7. The damage accumulation sequence has been investigated by time-lapse microCT. Are there any 3D visualization pictures or videos showing the damage sequence?

Yes. We first used real-time 3D X-ray imaging whilst subjecting the welded joints to tensile straining to observe the damage development from micro-void nucleation through growth to coalescence to final failure. Then, 3D volume renderings at different loading stages were acquired through image processing. Finally, these images were set as frames to form the animation using the Avizo software.

Action: *We have provided an animation named Supplementary Movie in the supplementary information.*

8. Change KAM maps in Fig.2c to GND density maps since the softening behavior depends on dislocation density. The crystal plasticity finite element simulation results are shown in Fig. S6. However, the detailed modelling is not provided, the constitutive model in particular.

Action: *As suggested, we have changed KAM maps to GND density maps in Fig. 2c where the GND density has been calculated using MTEX toolbox.*

We have added the constitutive model for crystal plasticity finite element modelling to the Methods section.

REVIEWERS' COMMENTS

Reviewer #1 (Remarks to the Author):

I am happy with the changes to the paper to address my concerns.
I am still unable to access the video file unfortunately as it is saying it is corrupted.

Reviewer #2 (Remarks to the Author):

The manuscript is now acceptable after careful revision.